# Anti-Inflammatory, Antioxidant, and Wound-Healing Properties of Cyanobacteria from Thermal Mud of Balaruc-Les-Bains, France: A Multi-Approach Study

**DOI:** 10.3390/biom11010028

**Published:** 2020-12-29

**Authors:** Justine Demay, Sébastien Halary, Adeline Knittel-Obrecht, Pascal Villa, Charlotte Duval, Sahima Hamlaoui, Théotime Roussel, Claude Yéprémian, Anita Reinhardt, Cécile Bernard, Benjamin Marie

**Affiliations:** 1UMR7245 MCAM MNHN-CNRS, Muséum National d’Histoire Naturelle, CP 39, 12 Rue Buffon, F-75231 Paris, CEDEX 05, France; justine.demay1@mnhn.fr (J.D.); sebastien.halary@mnhn.fr (S.H.); charlotte.duval@mnhn.fr (C.D.); sahima.hamlaoui@mnhn.fr (S.H.); theotime.roussel@mnhn.fr (T.R.); claude.yepremian@mnhn.fr (C.Y.); 2Thermes de Balaruc-Les-Bains, 1 Rue du Mont Saint-Clair BP 45, 34540 Balaruc-Les-Bains, France; anita.reinhardt@thermesbalaruc.com; 3CNRS, Université de Strasbourg, PCBIS Plate-Forme de Chimie Biologique Intégrative de Strasbourg UMS, 3286, F-67412 Illkirch, France; aobrecht@unistra.fr (A.K.-O.); pvilla@unistra.fr (P.V.); 4Labex MEDALIS, F-67000 Strasbourg, France

**Keywords:** cyanobacteria, thermal mud, natural products, anti-inflammatory, bioactivity

## Abstract

*Background:* The Balaruc-les-Bains’ thermal mud was found to be colonized predominantly by microorganisms, with cyanobacteria constituting the primary organism in the microbial biofilm observed on the mud surface. The success of cyanobacteria in colonizing this specific ecological niche can be explained in part by their taxa-specific adaptation capacities, and also the diversity of bioactive natural products that they synthesize. This array of components has physiological and ecological properties that may be exploited for various applications. *Methods:* Nine cyanobacterial strains were isolated from Balaruc thermal mud and maintained in the Paris Museum Collection (PMC). Full genome sequencing was performed coupled with targeted and untargeted metabolomic analyses (HPLC-DAD and LC-MS/MS). Bioassays were performed to determine antioxidant, anti-inflammatory, and wound-healing properties. *Results:* Biosynthetic pathways for phycobiliproteins, scytonemin, and carotenoid pigments and 124 metabolite biosynthetic gene clusters (BGCs) were characterized. Several compounds with known antioxidant or anti-inflammatory properties, such as carotenoids, phycobilins, mycosporine-like amino acids, and aeruginosins, and other bioactive metabolites like microginins, microviridins, and anabaenolysins were identified. Secretion of the proinflammatory cytokines TNF-α, IL-1β, IL-6, and IL-8 appeared to be inhibited by crude extracts of *Planktothricoides raciborskii* PMC 877.14, *Nostoc* sp. PMC 881.14, and *Pseudo-chroococcus couteii* PMC 885.14. The extract of the *Aliinostoc* sp. PMC 882.14 strain was able to slightly enhance migration of HaCat cells that may be helpful in wound healing. Several antioxidant compounds were detected, but no significant effects on nitric oxide secretion were observed. There was no cytotoxicity on the three cell types tested, indicating that cyanobacterial extracts may have anti-inflammatory therapeutic potential without harming body cells. These data open up promising uses for these extracts and their respective molecules in drugs or thermal therapies.

## 1. Introduction

Cyanobacteria, the so-called blue-green algae, are an ancient lineage of photosynthetic prokaryotes found worldwide. They are notorious for their harmful effects on human health because they produce toxins such as microcystins, anatoxins, and saxitoxins [1,2,3]. Not all species synthesize cyanotoxins, and cyanobacterial compounds such as pigments and specialized metabolites have been studied for their beneficial actions against cancer and as antioxidants and anti-inflammatories [4,5]. In recent years, the improvement of genome sequencing techniques and the increasing number of available genomes uncovered the great potential for specialized metabolite production by cyanobacteria [6,7,8,9]. Specialized metabolites, also known as secondary metabolites or natural products, are widespread in plants, fungi, and bacteria, but are especially abundant in cyanobacteria [10]. In cyanobacteria, genome exploration has revealed numerous biosynthetic gene clusters belonging to non-ribosomal pathways. These include non-ribosomal peptide synthetase (NRPS), polyketide synthetase (PKS), and members of other ribosomal pathways, such as ribosomally synthesized and post-translationally modified peptide (RiPPs), terpene, or alkaloid pathways [8,11]. Thanks to the diverse metabolic activity of cyanobacteria, their natural products have attracted a great deal of interest in research for new bioactive substances [12,13,14,15].

Benthic cyanobacteria especially have shown a wide range of specialized metabolites. This is illustrated by the remarkable number of compounds isolated from the *Lyngbya majuscula-Moorea producens* genus [4,9]. Because they are attached to a substrate, benthic cyanobacteria are subject to intense space competition and to stresses like desiccation, high light intensity, and predation by grazers that forced them to develop a large repertoire of chemical defences [16,17,18]. Therapeutic thermal mud from the thermal baths of Balaruc-les-Bains (France) has been used to treat various illnesses since ancient Roman times. For joint diseases such as osteoarthritis and rheumatism, the matured thermal mud (peloid) is applied directly to the patient’s skin. Two studies were conducted in the 1980 s to evaluate the specific algal communities colonizing this peloid, and both showed that diverse genera of microalgae and cyanobacteria have colonized the mud with a predominance of cyanobacteria belonging to the *Phormidium* genus [19,20]. It was hypothesized that compounds produced by this microflora, in association with heat and microelements in the hot spring water, were responsible for the beneficial therapeutic properties of Balaruc-les-Bains mud, as proposed for other thermal baths [21,22,23]. However, the thermal mud of Balaruc-Les-Bains still remains a relatively unexplored environment, particularly with regard to identifying the therapeutically active compounds produced by the microflora associated with the mud.

In 2014, a new isolation campaign was initiated to ascertain the evolution of the microbial communities in the maturing mud during the five-month summer period. Several species of cyanobacteria and microalgae were isolated and maintained in the Paris’ Museum Collection (PMC) [24]. Nine cyanobacteria, belonging to four different orders with different morphotypes (Table 1), were selected for assessment of metabolite production, as they belong to genera considered as prolific producers of potent bioactive compounds. 

To take advantage of the diverse bioactivities of the compounds produced by cyanobacteria, the anti-inflammatory, antioxidant, and wound-healing potential of the nine strains (Table 1) were investigated. Numerous compounds produced by these cyanobacteria have been described, including pigments, such as phycobilins, chlorophylls, carotenoids, and scytonemin, and specialized metabolites—mycosporine-like amino acids, aeruginosins, and honaucins [4,25]. The simple screening for activity as a way to identify new therapeutic metabolites has recently been greatly accelerated by improved genome sequencing methods and the development of genome mining to identify and characterize specialized metabolite biosynthetic gene clusters (BGCs) [26]. In consequence, the metagenomes of the present nine Balaruc cyanobacterial strains have been sequenced in-depth and biosynthetic genes for pigments and specialized metabolites have been characterized. Targeted pigment dose studies have been performed and untargeted LC-MS/MS investigations have been developed to determine the diversity of compounds produced by the nine strains. In parallel, crude water and methanol extracts have been tested for anti-inflammatory, antioxidant, wound-healing, and cytotoxic properties to evaluate their potential for further therapeutic development.

## 2. Materials and Methods 

### 2.1. Biological Material

Nine monoclonal cyanobacteria strains were isolated in 2014 from thermal mud in Balaruc-Les-Bains, France, and have been maintained as non-axenic cultures in the Paris Museum Collection (PMC) (Table 1) [24]. Sampling and taxonomy of the strains were described elsewhere [27]. The isolated cyanobacteria were grown in Z8 medium at 24 °C. During experiments, the cultures were maintained in 1-L Duran bottles with a photon flux density of 12 µmol·m^−2^·s^−1^ (PAR, photosynthetical active radiation) and a 16:8 h light:dark cycle. Cultures were transferred into fresh Z8 medium every six weeks to maintain them in the active growth phase.

### 2.2. Genome Sequencing and Assembly

After a centrifugation step of 150 mL cultures of each strain, DNA extractions were performed on the pellets using the ZymoBIOMICS DNA mini kit (Zymo Research, Irvine, CA, USA) following the manufacturer’s protocol. The Macherey Nagel genomic DNA and total RNA purification kit with NucleoBond AXG20 columns and buffer set III (Macherey-Nagel, PA, USA) was employed for PMC 881.14 and PMC 884.14. Metagenome sequencing was performed by GenoScreen (GenoScreen, Lille, France) using the Nextera XT DNA sample preparation kit (Illumina) for 2 × 250 bp and using a SMRT2 cell (PacBio) Illumina and Pacbio raw reads were corrected using SPAdes 3.12 and Canu 1.8, respectively, before the assembly performed with Unicycler hybrid-assembler, with default parameters [28,29,30]. For each new assembly, scaffolds were binned using MyCC (k-mer size = 4, minimal sequence size = 1000) and taxonomically annotated using CAT [31]. Congruent data between both methodologies allowed to characterize the cyanobacteria draft genomes.

### 2.3. In Silico Analyses

The genome assemblies were integrated on the MicroScope platform v3.14.1 (https://mage.genoscope.cns.fr/microscope/home/index.php) [32]. Biosynthetic gene clusters (BGCs) for specific metabolites were identified using the antiSMASH v5.0.0 [33] and MIBiG v1.4 [34] software available through the MicroScope platform. These tools allowed characterization of the gene cluster organization and predicted the molecules potentially produced by comparison of homologies between the query and the BGC sequences annotated in the databases. For each BGC identified, we carefully checked the automatic prediction against the literature on the biosynthetic pathways described to obtain a “curated” annotation of the detected clusters with the highest confidence possible. As the automatic annotation tools did not detect pigment biosynthetic genes, a targeted approach was performed using BLASTp and the search-by-keywords function of the MicroScope platform.

### 2.4. Pigment Composition Analysis

The pigment composition of the strains was determined in triplicate from 10 mL of culture filtered through Whatman GF/F filters (⌀0.7 µm) and freeze dried. For extracting lipid-soluble pigments like carotenoids and chlorophylls, filtered biomasses were extracted with methanol, pooled, and quantified by HPLC-DAD (as described in Ras et al., 2008) [35]. For determination of the water-soluble phycobiliprotein pigments, filtered biomasses were analyzed following a published procedure [36].

### 2.5. Metabolite Extraction, Analysis by Mass Spectrometry, and Annotation

For each cyanobacterial strain, biomasses from five successive subcultures (5 × 500 mL) were collected at the end of the exponential growth and centrifuged at 3220× *g* for 10 min. The pellets were lyophilized (Freezone 2.5 L, Labconco, Italia) and pooled for metabolite extraction. Two different metabolite extractions were performed with the same pooled samples (200 mg dried weight) to increase the likelihood of identifying all the metabolites: (1) a mild hydrophobic single extraction with MeOH/water, 75%/25%, *v*/*v* (solvent A) and (2) a hydrophilic single extraction with ultrapure water acidified with 0.1% formic acid (solvent B). Lyophilized cell pellets were weighed and 1 mg in 100 µL of the extraction solution was sonicated (SONICS Vibra Cell, 130 Watt, 20 Khz) at 80% power three times for two minutes each time (30 s rest between). Prior to analysis, the sonicated extracts were centrifuged at 12,000× *g* for 10 min at 4 °C. Four microliters aliquots of the supernatants were run on an ultra-high-performance liquid chromatograph (UHPLC Ultimate 3000, Thermo, Waltham, MA, USA) using a Polar Advances II 2.5 pore C18 column (Thermo, Waltham, MA, USA) at a 300 µL·min^−1^ flow rate with a linear gradient of acetonitrile in 0.1% formic acid (5–90% for 21 min) coupled with a high-resolution mass spectrometer. Chromatography runs were done in triplicate. The eluted metabolites were analyzed using an electrospray ionization hybrid quadrupole time-of-flight (ESI-QqTOF) high-resolution mass spectrometer (Compact Bruker, Bremen, Germany) in positive simple MS or positive autoMS/MS mode with information-dependent acquisition (IDA), on the 50 to 1500 *m*/*z* range at 2 Hz or between 2 and 8 Hz speed, for MS and MS/MS, respectively, according to relative intensity of parent ions, in consecutive cycle times of 2.5 s, with an active exclusion of previously analysed parents. The data were analyzed with the DataAnalysis 4.4 software for internal recalibration (<0.5 ppm) and MGF exports were generated from MS/MS spectra between 1 and 15 min. The raw MS and MS/MS data were analyzed with MetaboScape 4.0 software (Bruker, Bremen, Germany) to automatically search and group together all classical adduct forms ([M + H]^+^, [M + 2H]^+^, [M + 3H]^+^, [M + Na]^+^, [M + K]^+^, and [M + NH_4_]^+^) using a threshold value of 0.8 for the co-elution coefficient factor. Metabolite annotation was attempted according to the precise mass of the molecules and their respective MS/MS fragmentation patterns with regards to MS/MS libraries (NIH, GNPS, EMBL, MassBank, ReSpect, etc.) and the CyanoMet database [37] of >2100 cyanobacterial metabolites. Results were confirmed by running 36 commercially available standards of specific cyanobacterial metabolite families: cyanopeptolins, aeruginosins, microginins, anabaenopeptins, aerucyclamides, microcystins, saxitoxins, anatoxins, and cylindrospermopsins.

### 2.6. Molecular Networking

MetGem v1.2.2 [38] was used to create a molecular network from the MS/MS data (in mgf format) for two out of the nine extracts. The network was created with edges filtered to have a cosine score above 0.65 and more than four matched peaks. Further edges between two nodes were retained in the network only if each of the nodes appeared in each of the other’s top ten most similar nodes, respectively. The MetGem database search function was used to screen each spectrum against GNPS and ISDB spectral libraries.

### 2.7. Bioassays

Activity assays were performed using crude extracts from the same nine cyanobacteria as for metabolomic analysis and the same method with solvents A and B. Methanol extracts were evaporated with a RotaVap R-210 (Büchi, Flawil, Switzerland) and aqueous extracts were lyophilized. Each was then resuspended in DMSO/water (50%/50%) at a concentration of 30 mg·mL^−1^ (dried extract). 

The antioxidant potential of the crude extracts was assessed by measuring nitric oxide (NO) production by the macrophage-like cell line, RAW 264.7, stimulated with 5 µg·mL^−1^ lipopolysaccharide (LPS, Sigma L6636). Quercetol was used as the positive antioxidant control and results were expressed as percent inhibition compared to the maximal NO level induced by LPS in 0.1% DMSO [39]. Following 24 h of treatment, NO production was determined indirectly with quantification of nitrite (NO_2_^−^) accumulation in culture media using Griess Reaction. Briefly, 100 µL of Griess reagent (Invitrogen G-7921) was added to 100 µL of cellular supernatant samples into 96-well plates (Greiner 655101) and left 30 min at room temperature under shaking. Absorbance at 540 nm was measured using an Envision (Perkin Elmer) microplate reader. The reduction or increase in nitrite levels was calculated by the percentage of absorbance of sample relative to the positive control (5 µg·mL^−1^ LPS with 0.1% DMSO).

Anti-inflammatory activity of the extracts was evaluated by determining their effect on the concentration of the proinflammatory cytokines, TNFα, IL-1β, IL-6, and IL-8, in human peripheral blood mononuclear cells (PBMCs). PBMCs were stimulated with 5 µg·mL^−1^ LPS and the extracts were then added. Cytokine concentration was then measured using HTRF technology with fluorescent antibodies [39]. Dexamethasone was used as positive control and results were expressed as percent inhibition relative to the maximal cytokine secretion induced by LPS.

Cell migration assays were performed using the HaCat cell line of human keratinocytes. Extracts were pipetted onto HaCat monolayers previously wounded and treated to prevent cell proliferation. The plate was then placed into the IncuCyte^TM^ (Essen Bioscience, Ann Arbor, MI) live-cell imaging instrument to measure the cell migration onto the wounded area for 48 h. Fetal bovine serum (FBS) was used as positive control as it enhances cell migration onto the wounded area [40].

Cell viability was also measured during the three previous tests with the WST-1 assay (Ozyme, Saint-Quentin-en-Yvelines, France) following protocol described previously [41] in order to validate the bioassay results and to assess the potential cytotoxicity of the crude extracts. For this test, an extract was considered cytotoxic if the cell viability was less than 70% [39].

## 3. Results

### 3.1. Characteristics of the Cyanobacterial Genomes

Results of analysis of the nine cyanobacterial genomes are presented in Table 2. High CheckM completeness scores (>97.1%) indicate that the genomes were mostly complete and well assembled. A low contamination score (<1.4%) was found for all genomes, which means a low risk of potential contamination of assemblies with exogenous sequences. The mean size of the genomes was 7.8 Mb, ranging from 5.8 Mb for the *Pseudo-chroococcus* strain, PMC 885.14, to 13.2 Mb for the *Calothrix* sp., PMC 884.14. An important difference was observed between the numbers of contigs for each strain, which ranged from five contigs for *L. boryana,* PMC 883.14, to 648 for *Nostoc* sp., PMC 881.14.

### 3.2. Pigment Composition

Pigments are some of the primary metabolites that exhibit potent antioxidant and/or anti-inflammatory properties. Thus, we investigated the biosynthetic pathways of the phycobiliproteins, scytonemin, carotenoids, and chlorophylls in the nine strain genomes (Table 3). 

Phycocyanin pathways were identified in all nine cyanobacteria, while the phycoerythrin genes were only found in *P. raciborskii* PMC 877.14, *Nostoc* sp. PMC 881.14, *Calothrix* sp. PMC 884.14, and *Pseudo-chroococcus couteii* PMC 885.14. Genes for the yellow-brown pigment scytonemin were detected in the genomes of PMC 881.14, *Aliinostoc* sp. PMC 882.14, and PMC 884.14. All the carotenoids biosynthetic genes were identified in *Laspinema* sp. PMC 878.14, *L. martensiana* PMC 880.14, *Aliinostoc* sp. PMC 882.14, *L. boryana* PMC 883.14, and *Calothrix* sp. PMC 884.14. *P. raciborskii* PMC 877.14 lacked the *crtO* gene, suggesting that the strain could not produce echinenone and ketocarotenoid derivatives. The *crtG* gene was not detected in PMC 877.14, PMC 879.14, PMC 881.14, or PMC 885.14, which means that the strains do not have the ability to produce the hydroxylated carotenoids caloxanthin and nostoxanthin, or the myxoxanthophyll 2-hydroxymyxol 2′-methylpentoside.

Spectrophotometric analysis and HPLC were used to confirm pigment production (Table 4). Phycoerythrin was detected in the four strains in which the biosynthetic genes were identified. The concentration of phycoerythrin in *P. raciborskii* PMC 877.14 was 33.3 µg·mg^−1^ (dried weight, DW) compared to 9.9 µg·mg^−1^ DW for *Calothrix* sp. PMC 884.14, 3.3 µg·mg^−1^ DW for *P. couteii* PMC 885.14 and 2.8 µg·mg^−1^ DW for *Nostoc* sp. PMC 881.14. The highest concentration of phycocyanin was extracted from *P. raciborskii* at 32.5 µg·mg^−1^ DW followed by *L. boryana* at 24.9 µg·mg^−1^ DW and *Aliinostoc* sp. at 21.3 µg·mg^−1^ DW). The phycocyanin concentration in extracts of the other strains ranged from 18.4 µg·mg^−1^ DW in *Laspinema* sp. to 3.5 µg·mg^−1^ DW in *Pseudo-chroococcus couteii*. The largest allophycocyanin producers were *Aliinostoc* sp. and *L. boryana,* with 15.6 and 15.2 µg·mg^−1^ DW extracted, respectively. The allophycocyanin concentration in extracts of the other strains was lower, ranging between 7.9 µg·mg^−1^ DW for *P. raciborskii* and 2.7 µg·mg^−1^ DW for the *P. couteii* strain. 

The amount of chlorophylls extracted was also quite variable among the different strains. The largest amount, 35.2 µg·mg^−1^ DW, was extracted from *P. raciborskii,* while the lowest was from *Nostoc* sp. PMC 881.14 at 1.5 µg·mg^−1^ DW.

Three different carotenoids were detected in the nine extracts. β-carotene was identified in all the strains, with concentrations ranging from 3.65 µg·mg^−1^ DW for *P. raciborskii* to 0.46 µg·mg^−1^ DW for *Calothrix* sp. and *L. martensiana*. Zeaxanthin was detected in extracts of *P. raciborskii*, *Laspinema* sp., *M. vaginatus*, *L. martensiana*, *Aliinostoc* sp., *L. boryana*, and *P. couteii*. Myxoxanthophyll-like pigments were found in six strains, but not in *M. vaginatus*, *L. boryana*, and *P. couteii*. 

### 3.3. Specialized Metabolites 

Following the analysis of the nine assembled genomes, 124 specialized metabolite biosynthetic gene clusters (BGCs) were retrieved (Table 5 and Appendix A). The numbers of BGCs per strain varied from 7 to 27. Among these 124 BGCs, only 31 were correctly predicted to have the molecule produced by the cluster, representing 24.4% reliable BGC annotation (Table 5, “Identified Clusters” column). The largest number of annotations was obtained for *Aliinostoc* sp. PMC 882.14 and *Calothrix* sp. PMC 884.14, in which the production of six and five metabolites, respectively, was accurately predicted. Among the nine strains, those belonging to the Nostocales order have been shown to present the highest number of BGCs compared to other strains (Table 5). This observation is consistent with the work of Shih and co-workers who showed that heterocytic cyanobacteria tend to exhibit a higher number of BGCs [7]. Automatic annotations accurately predicted the phytoene biosynthetic pathway in all nine cyanobacterial strains, and the pathways of the mycosporine-like amino acids shinorine and mycosporine-glycine for *Aliinostoc* sp. PMC 882.14 and *Calothrix* sp. PMC 884.14. The other BGCs exhibited mostly inaccurate predictions (see Appendix A for automatic prediction results and detailed organization of the BGCs).

Expert manual annotation also allowed the identification of a schizokinen-like biosynthetic cluster in *L. martensiana* PMC 880.14, from one BGC automatically annotated as siderophore. Heterocyst glycolipid BGCs were identified in the three Nostocales strains (*Nostoc* sp. PMC 881.14, *Aliinostoc* sp. PMC 882.14 and *Calothrix* sp. PMC 884.14). The BGC of ribosomally synthesized peptides with post-translational modifications (RiPPs) of cyanobactins was identified in two strains: *Laspinema* sp. PMC 878.14 and *Calothrix* sp. PMC 884.14. Another RiPP, microviridin, was characterized from *Aliinostoc* sp. PMC 882.14, and shared strong homology with the microviridin K biosynthetic gene cluster. The type I polyketide synthetase (PKS) of amycomicin has been proposed for a PKS gene cluster from *Aliinostoc* sp. PMC 882.14. The cylindrocyclophane-like gene cluster has been identified from an NRPS cluster in *Calothrix* sp. PMC 884.14. Our manual expert annotation also allowed the identification of a microginin BGC from a hybrid NRPS-PKS automatically annotated as the puwainaphycin cluster in *Aliinostoc* sp. PMC 882.14. No cyanotoxin (microcystins, anatoxins, cylindrospermopsins, saxitoxins, etc.) gene clusters were identified in any of the strains. Among the 93 unknown clusters, 51 showed homologous matches with clusters identified in other strains, without characterization of the molecule synthesized, while 42 were fully uncharacterized.

The metabolites putatively identified by the genomic approach were also subjected to LC-MS/MS analysis. A dataset of 3986 analytes was obtained, with their respective intensity in the different extracts (water or methanol) and their respective fragmentation pattern. The molecular network assembled from the analyzed MS/MS is presented in Figure 1 and Appendix A. This type of representation makes it possible to group together analytes in clusters that sharing similarities in their fragmentation spectra and therefore have structural similarities. In these figures, one node represents one analyte with its own mass and fragmentation spectra. The molecular networking performed with the GNPS algorithm allowed us to group 1155 analytes (29%) into 128 clusters containing at least three different analytes. The remaining analytes were classified in pairs or singletons. In Figure 1, nodes have been colored according to the strain from which they were identified. Appendix A represents the same analysis while nodes have been colored according to their original fraction (water or methanol). Most of the analytes were observed in several strain extracts (Figure 1A) while some were identified exclusively in a single strain (Figure 1B). The same pattern was identified at the cluster level, with several clusters containing multi-species analytes (Figure 1C) and almost 50 strain-specific clusters (Figure 1D). Comparisons with public and in-house spectral MS/MS databases together with the molecular formula database of more than 2100 cyanobacterial metabolites (CyanoMetDB) allowed the prospective annotation of 193 of those analytes (Figure 1 and Appendix A). Primary metabolites have most often been identified, including diverse dipeptides, ribonucleosides, and cofactors (NAD+), as well as glycerolipids, polysaccharides, lysophosphatidylcholines and lysophosphoethanolamines. Several mycosporine-like amino acids (MAAs) were also characterized with specific matches to shinorine, mycosporine-glycine and palythine. A few specialized cyanobacterial metabolites were identified, notably aeruginosin-like compounds, anabaenolysins, largamides, microphycins, microviridins, and microginins. According to the propagation principle, one would expect that analytes from the same clusters should belong to the same chemical family. 

The annotation details for the targeted list of metabolites of interest from the nine cyanobacterial strains, selected for their potential bioactivities, are given in Table 6. Cyanobacterial pigments are well known for their antioxidant and/or anti-inflammatory properties, and several metabolites with potentially similar bioactivities were identified. Several different MAAs, known to be antioxidant metabolites, were detected in seven strains. Mycosporine-glycine was annotated for PMC 882.14 and PMC 884.14. Shinorine has been detected in *Aliinostoc* sp. PMC 882.14, implying a remarkable antioxidant potential for these two cyanobacteria. One congener of the anti-inflammatory aeruginosin family, aeruginosin TR642, was also annotated in *L. boryana* PMC 883.14, highlighting the anti-inflammatory potential of this strain.

### 3.4. Bioactivity Assessment

Specific bioassays were done to determine the antioxidant, anti-inflammatory and wound healing activities of the cyanobacterial extracts. Both aqueous and methanol extracts were tested at concentrations of 1, 5, 10, and 50 µg·mL^−1^ (Table 7). 

Antioxidant activity was evaluated by measuring the effect of crude extracts on the amount of the reactive oxygen species, nitric oxide (NO) produced by the murine macrophage cell line, RAW 264.7, subjected to oxidative stress by exposure to lipopolysaccharides (LPS). NO is evolved during normal oxidative metabolism, but an excess of free radicals can cause cell damage and overwhelm protective antioxidant enzymes. A positive antioxidant effect was recorded when an extract reduced the amount of NO produced comparable to the negative control. No significant inhibition of NO formation was seen at the extract concentrations tested. A slight increase in NO production (+20%) was observed for the aqueous extract of *Nostoc* sp. PMC 881.14 at 50 µg·mL^−1^ (Table 7), suggesting that this extract might have pro-oxidant capabilities.

The anti-inflammatory properties have been evaluated by measuring specific cytokine secretion by immune cells (PBMCs) exposed to LPS. The secretion of four cytokines, involved in upregulation of the inflammatory response, were assayed: TNF-α, Il-1β, IL-6, and IL-8 [42,43]. Several aqueous or methanol extracts showed effects on cytokine release, with the majority occurring at the higher concentrations of 10 µg·mL^−1^ DW and 50 µg·mL^−1^ DW (Table 7). Some of the responses were proinflammatory with an increase in cytokine production, while others showed anti-inflammatory activity and decreased cytokine secretion. Distinct dissimilarities in the cytokine responses were observed in several strains, notably for the aqueous extracts of *Aliinostoc* sp. PMC 882.14 that enhanced IL-1β secretion and inhibited Il-6 secretion at 50 µg·mL^−1^. The methanol extract of *P. raciborskii* PMC 877.14 and the aqueous extract of *P. couteii* PMC 885.14 had the best anti-inflammatory properties as they each simultaneously inhibited the release of two cytokines involved in proinflammatory responses—TNF-α and Il-1β, and Il-1β and Il-6 respectively. The greatest inhibition of cytokine secretion was observed with the aqueous extract of PMC 877.14 with a 57% decrease of IL-1β, compared to the negative control.

The wound-healing activity of the crude cyanobacterial extracts was also evaluated by monitoring the cellular migration of the keratinocyte cell line, HaCaT, for 48 h following injury to the monolayer. The aqueous extract of *Aliinostoc* sp. PMC 882.14 at 10 µg·mL^−1^ slightly increased keratinocyte migration by 22% ± 7 compared to the negative DMSO control, suggesting a beneficial effect of this extract on wound healing (Table 7).

Cell viability was determined on the three cell types used in the bioassays: murine macrophages (RAW 264.7), human keratinocytes (HaCaT), and peripheral blood mononuclear cells (PBMC) (Appendix A). This assay has allowed us to assess the viability of the respective cell types used in the bioassays, as well as to evaluate the potential cytotoxicity of the crude extract on these different cell types. No loss of viability of HaCaT human cell line and human PBMC was observed. Murine macrophages RAW 264.7 were apparently more sensitive and showed a slight loss of viability.

## 4. Discussion

### 4.1. Genome Quality, Genome Mining, and Molecular Network Analysis

Application of the genome mining approach led to the identification and characterization of biosynthetic gene clusters (BGCs) for natural products. This approach requires high-quality genomic data to confidently annotate the BGCs and excludes doubtful affiliations, especially in the context of genome sequencing and assembly with meta-genomes generated from nonaxenic strains [44]. The median size of the nine cyanobacterial genomes was 7.3 Mb, which is higher than the 5.6 Mb median size of the cyanobacterial genomes referenced in GenBank [45]. However, Alvarenga and co-workers [46] showed that the latter value was biased by the high number of *Prochlorococcus* genomes available wherein the size was < 3 Mb. The GC percentage of our nine genomes (42%) was comparable to reference cyanobacterial genomes, already available on public database such as NCBI’s. 

The high completeness scores (>97%) obtained for the nine genome assemblies led us to conclude that the draft genomes for all strains were nearly, if not totally, complete. The low contamination score (<1.4%) indicates that our double binning strategy, which was based on both BLAST and K-mer contig clustering, gives reliable results for obtaining uncontaminated cyanobacterial genomes from nonaxenic monoclonal cultures. According to Parks and co-workers, who developed the CheckM tool, the values of these two scores indicate high quality for the nine cyanobacterial genome assemblies obtained from metagenomic data [47]. Automated tools, such as antiSMASH, together with experts’ participation, allowed the annotation of 24% of the 124 identified BGCs. BGCs without annotations were considered orphan gene clusters for which the synthesized metabolite was unknown. However, 52% of these orphan clusters had high homologies with other orphan clusters already observed in other cyanobacteria. This opens up the possibility of global comparative studies of these different strains characterizing their metabolites, as was done for the cyanobactin biosynthesis pathway [48].

### 4.2. Cytotoxic Properties 

We investigated the potential cytotoxicity of the different extracts on the RAW 264.7, HaCat cell lines and on human PBMCs. No significant cytotoxicity was detected on the two last cell types while a slight loss of viability was observed on the more sensitive murine cell line (RAW 264.7) with some extracts. Overall, no cyanotoxins, such as microcystins, anatoxins, saxitoxins, cylindrospermopsins, or nodularins, were identified in the crude extracts and no cyanotoxin BGCs were detected in the nine genome assemblies, suggesting that the nine strains did not present a toxicity risk and are suitable for therapeutic purposes. However, two potential cyanobactin BGCs were identified in *Laspinema* sp. PMC 878.14 and *Calothrix* sp. PMC 884.14. The two clusters represented one gene encoding a heterocyclase enzyme. This may mean that the cyanobactins produced by these strains belong to the group of heterocyclized cyanobactins, which includes patellamides, tenuecyclamides, trichamide, aeruginosamide, and others [11] with potentially cytotoxic activities [49,50]. However, not all cyanobactins exhibit cytotoxic activity and no known cyanobactin compounds were identified in crude extracts of the two strains.

### 4.3. Antioxidant Properties

Antioxidants have been studied for many years as novel treatments for diseases such as atherosclerosis, osteoarthritis, and neurodegenerative disorders and have been generally recognized as compounds that promote well-being [51]. Pigmented compounds are produced by all cyanobacteria and have shown potentially beneficial properties [52,53,54]. Biosynthetic genes for phycocyanins and chlorophylls have been detected in the nine cyanobacteria, but their relative amounts varied considerably among the different strains. *Calothrix* sp. PMC 884.14, *L. martensiana* PMC 880.14, *P. couteii* PMC 885.14, and *Nostoc* sp. PMC 881.14 possessed a thick sheath and abundant mucilage that may have affected the pigment extraction efficiency or have biased the measurement of pigment mass by increasing the apparent lyophilisate weight or reducing the number and volume of extracted cells. These observations are consistent with the conclusions of Finkel and co-workers who demonstrated phylogenetic difference in macromolecular composition, notably of chlorophyll *a* [55]. Phycoerythrin genes have been identified in *P. raciborskii* PMC 877.14, *Nostoc* sp. PMC 881.14, *Calothrix* sp. PMC 884.14, and *P. couteii* PMC 885.14. Phycoerythrin was detected and quantified in these four strains and appeared to be produced constitutively. Carotenoids pathways have also been characterized in the nine assemblies according to the pathways proposed by Kosourov et al. (2016). These authors described sixteen genes coding for enzymes involved in the synthesis of hydroxylated carotenoids, ketocarotenoids, echinenone, and myxoxanthophylls [56]. Of the known carotenoids described in cyanobacteria [57,58], only four have been identified in this study: zeaxanthin, canthaxanthin, myxoxanthophyll-like molecules, and β-carotene (Table 4). Only this last has been detected in all nine strains. Regarding the others, several reasons could explain their non-detection. Some strains did not possess all the genes necessary for the synthesis of certain carotenoids. This is the case for *P. raciborskii* PMC 877.14 which lacks the *crtO* gene encoding a β-carotene ketolase required for canthaxanthin synthesis. This enzyme catalyzes the transformation of β-carotene into echinenone and echinenone into canthaxanthin; thus, this strain is unable to produce canthaxanthin. For the strains possessing all carotenoid biosynthetic genes but appearing to lack carotenoid, it is possible that the amount extracted from the strains is below the detection limit or that some regulatory processes for intermediary products in the linear pathway are inhibiting carotenoid biosynthesis. 

Specialized metabolites with potential antioxidant properties have also been detected in the cyanobacterial extracts. Mycosporine-like amino acids (MAAs) are a family of small metabolites identified in fungi, plants, and bacteria, including cyanobacteria, and showing antioxidant and photo-protective activities [59]. BGCs for MAAs have been characterized by automatic annotation tools from *Aliinostoc* sp. PMC 882.14 and *Calothrix* sp. PMC 884.14. Both BGCs contained one demethyl-4-deoxygadusol synthase, one O-methyltransferase, one ATP-grasp domain and one NRPS module described for synthesis of MAAs [60,61]. Both strains produced mycosporine-glycine, the first molecule synthesized from the 4-deoxygadusol precursor. Only PMC 882.14 had the ability to produce shinorine, as this metabolite requires serine incorporation brought about by one adenylation domain. Manual exploration of the *Nostoc* sp. PMC 881.14 genome identified a MAAs gene cluster containing one D-Ala-D-Ala ligase replacing the NRPS gene. This gene cluster organization was similar to that described for *Cylindrospermum* stagnale PCC 7417 [61,62]. While 4-deoxygadusol (4-DG) in PMC 881.14 was synthesized from the shikimate pathway instead of the pentose phosphate pathway, mycosporine-ornithine was detected as in PCC 7417.

Unexpectedly, no effects on nitric oxide production were observed for any of the strains despite the antioxidant potential of their pigments and specialized metabolites (Table 6). This absence of effect could be due to low concentration of these pigment molecules in the crude extracts or the potential antagonistic effect of specific inhibitors. Cyanobacteria could also produce oxidative compounds as found for the *Nostoc* sp. PMC 881.14 aqueous extract, in which the cellular oxidative potential increased 20%. This increased NO formation could be caused by the lipopolysaccharide (LPS) content of this strain, even though cyanobacterial LPS is said to be a weak inducer of oxidant activity compared to other Gram-negative bacteria [63,64].

### 4.4. Wound-Healing Properties

Cell migration is involved in many diseases such as for metastasis and the development of atherosclerosis and arthritis, but is also necessary for normal physiological processes like keratinocyte migration in wound healing or the translocation of leukocytes in the inflammatory response and immune functions [65,66]. The *Aliinostoc* sp. PMC 882.14 extract stimulated slightly keratinocyte migration on the HaCaT cell line. Several cyanobacterial extracts have been studied in recent years for their potential wound-healing effects [67,68], but the mechanism of action of most of these compounds remains unclear. The aqueous extract of *Aliinostoc* sp. PMC 882.14 exhibited a wound healing effect, but did not promote Il-6 secretion. We hypothesized that this mechanism is different from the process observed in *Synechococcus elongatus* PMC 7942 [69], but could be more similar to pathways described for apratyramide, as it induced the secretion of growth factor [70]. Apratyramide, isolated from *Moorea bouillonii*, is a linear depsipeptide of five residues with two terminal N-methylated tyrosines. This terminal organization is quite similar to that of the microginin family [71] and a microginin BGC was identified and several congeners characterized by molecular networking in *Aliinostoc* sp. PMC 882.14 (Figure 1) suggesting the possibility that microginins could be involved in the observed wound healing activity as they share the same tyrosine pattern as apratyramide. Microginins have also been described as protease inhibitors [72,73,74] and angiotensin converting enzyme inhibitors [75,76]. Phlebitis and other vascular diseases have been cured in the thermal baths of Balaruc-les-Bains since ancient times, and the microginins produced by *Aliinostoc* sp. PMC 882.14 could be involved in these therapeutic properties.

### 4.5. Anti-Inflammatory Properties

Inflammation is a normal immune process involving a complex network of cells, regulatory pathways and pro- and anti-inflammatory molecular activities to balance the immune response. Defects in immunoregulation can lead to chronic inflammatory diseases such as rheumatoid arthritis, psoriasis, multiple sclerosis, and others [77,78]. Proinflammatory cytokine responses to the crude extracts showed different profiles depending on the strain tested (Table 7). Some extracts showed complex patterns with increasing secretion of one cytokine concomitant with decreased release of another cytokine, making it difficult to draw conclusions about their respective global anti-inflammatory activity. Nevertheless, three strains showed promising anti-inflammatory effects by inhibiting the production of one or more cytokines. Both methanol and water extract of *P. raciborskii* PMC 877.14 showed decreased levels of TNF-α and Il-1β. As PMC 877.14 is the most important phycocyanin producer of the nine strains from Balaruc-les-Bains, anti-inflammatory properties of the aqueous extract could be attributed to the phycobilin pigment. The aqueous extract of *P. couteii* PMC 885.14 inhibited the secretion of Il-1β and Il-6. As only low levels of phycocyanin were extracted from this strain, it is likely that another molecule is responsible for this activity. In a recent paper on the therapeutic properties of *Phormidium* sp. ETS05 from thermal mud in Italy, Zampieri et al. reported that hydrosoluble exopolysaccharides demonstrated anti-inflammatory properties without toxicity [23]. The same macromolecules could be responsible for the anti-inflammatory properties of *Pseudo-chroococcus couteii*, which possesses a thick polysaccharide sheath [27].

Scytonemin BGC has been detected in PMC 881.14. Scytonemin is a lipid-soluble, yellow-brown pigment with anti-inflammatory activity [79]. Unfortunately, we were not able to detect scytonemin or its congeners by our metabolomic analysis and this absence, or non-detection, has been attributed to unfavorable culture conditions for scytonemin production, as scytonemin synthesis has been shown to be upregulated by UV-A exposure [60,79]. As scytonemin could not be responsible for the anti-inflammatory properties of methanol extracts of PMC 881.14 and PMC 877.14, we hypothesized that other molecules must be responsible for the anti-inflammatory properties. Potential candidates could be lipid-derived products as they represent the biggest cluster in our network. This explanation is supported by research from Bruno et al. (2005) who described the anti-inflammatory activities of several galactosyl-diacyl-glycerol compounds [80].

### 4.6. General Comments

The various results obtained in the course of this study open up interesting perspectives in different fields of application and provide insights into the therapeutic mechanisms of thermal mud. In fact, it has been shown that both the “inorganic” and “organic” parts of the peloids confer therapeutic properties to thermal cures [81,82]. To go further in understanding these mechanisms for the thermal mud of Balaruc-les-Bains specifically, it would be interesting to study the metabolome of the strains and their bioactivities under other conditions, including co-culture experiments with the mud/thermal water mixture. This experiment would make it possible to mimic the natural maturation conditions of the peloid and would make it possible to study the whole metabolome and bioactivities of the thermal mud flora as has been done for other thermal baths [83,84]. Moreover, several studies have shown that the metabolome of cyanobacteria varies under co-culture conditions [85,86], allowing the hypothesis that bioactivities (including the antioxidant activity) could be enhance by synergistic interactions.

While the cyanobacterial extracts did not show cytotoxicity to the cell’s types tested, it might be advisable that we tested them against several other cell lines. Because of their transporters, some cell lines could be more sensible to cyanobacterial peptides, which need to be transport into the cell to be toxic. This is the case, for example, of microcystins, which are internalized by liver cells thanks to transporters specific to certain cellular tissues [87].

Beyond the applications sought for balneology, the metabolites identified within the framework of this study could potentially be used in other fields such as cosmetics or pharmaceuticals. It would therefore be interesting to go further in the characterization of the structures and the bioactivities of the whole metabolites. Indeed, cytotoxic compounds, such as cyanobactins or cylindrocyclophanes, have to be investigated in order to develop anticancer drugs when protease-inhibiting molecules (such as microviridins and microginins) can be used in the treatment of vascular diseases, hypertension, or infectious diseases [4].

For future experiments, it could be interesting to purify certain compounds and characterize their structures by HPLC and NMR spectrometry, to characterize their individual activities by other bioassays (other antioxidant tests, protease inhibition tests) and to follow their production kinetics during the different growth phases of the cyanobacteria strains of interest.

## 5. Conclusions

Cyanobacteria are prolific producers of natural compounds exhibiting a wide range of chemical structures and activities [4,5]. Although their biological functions remain largely unknown, some compounds have demonstrated their usefulness in pharmacology such as dolastatin 10 for the treatment of lymphoma in Hodgkin’s disease [15,88,89,90]. Here, we investigated the potential beneficial activities of nine cyanobacteria isolated from the thermal mud of Balaruc-les-Bains, France, in terms of the production of antioxidant and/or anti-inflammatory compounds using genomic, metabolomic, and bioactivity analyses. With only 24% of the biosynthetic gene clusters annotated, Balaruc’s cyanobacteria constitute a rich source of potentially valuable natural products. The nine strains, which do not express harmful cyanotoxins and did not show cytotoxicity against two human cell types, exhibited beneficial anti-inflammatory and wound-healing properties possibly arising from the identified pigments and specialized metabolites. Although no antioxidant activity was detected, strong anti-inflammatory potential was proven for *Planktothricoides raciborskii* PMC 877.14, *Nostoc* sp. PMC 881.14, and *Pseudo-chroococcus couteii* PMC 885.14, and a slight wound-healing function was detected in extracts from *Aliinostoc* sp. PMC 882.14. Antioxidant properties still need to be studied as several strains showed high production of antioxidant pigments, while our assays failed to detect activity. Nevertheless, our study shows a promising potential in further developments for Balaruc’s cyanobacteria and opens avenues of future work for applications of the active biomolecules, improvement of their production, and the resolution of their mechanism of action.

## Figures and Tables

**Figure 1 biomolecules-11-00028-f001:**
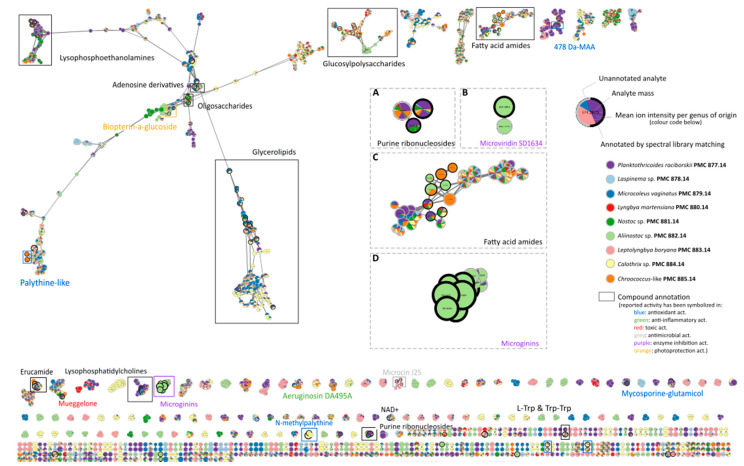
Molecular network of the metabolites produced by nine cyanobacteria isolated from Balaruc’s thermal mud. Annotations obtained by untargeted analysis are figured inboxes. Nodes have been colored according to the strain in which they have been identified. (**A**) Purine ribonucleosides cluster; (**B**) Microviridins cluster; (**C**) Fatty acid amides cluster; (**D**) Microginins cluster.

**Table 1 biomolecules-11-00028-t001:** List of cyanobacterial strains isolated from the Balaruc thermal mud and their corresponding strain reference numbers. PMC: Paris’ Museum Collection.

Order	Species	Strain Number	BioProject Accession Numbers
Chroococcales	*Pseudo-chroococcus couteii*	PMC 885.14	PRJNA686263
Synechococcales	*Leptolyngbya boryana*	PMC 883.14	PRJNA686260
Oscillatoriales	*Planktothricoides raciborskii*	PMC 877.14	PRJNA686238
	*Laspinema* sp.	PMC 878.14	PRJNA686242
	*Microcoleus vaginatus*	PMC 879.14	PRJNA686244
	*Lyngbya martensiana*	PMC 880.14	PRJNA686257
Nostocales	*Nostoc* sp.	PMC 881.14	PRJNA686258
	*Aliinostoc* sp.	PMC 882.14	PRJNA686259
	*Calothrix* sp.	PMC 884.14	PRJNA686262

**Table 2 biomolecules-11-00028-t002:** Characteristics of assembled genomes of the nine cyanobacteria isolated from thermal mud in Balaruc-Les-Bains obtained from the MicroScope platform. CheckM analysis has been performed for each genome using reference genomes (65–89 for 520–584 lineage-specific markers). CDS: Coding sequences.

Genus/Species	Strain	Genome Length (bp)	GC Content	Number of Contigs	Number of CDS	Number of rRNA	Average CDS Length (bp)	CheckM Completeness	CheckM Contamination
*P. raciborskii*	PMC 877.14	7,392,957	44%	15	7130	12	841.8	99.9 %	0.89 %
*Laspinema* sp.	PMC 878.14	7,343,542	47%	69	5829	9	1 022.4	99.3 %	1.16 %
*M. vaginatus*	PMC 879.14	6,904,242	46%	34	6400	6	904.5	99.6 %	0.22 %
*L. martensiana*	PMC 880.14	6,457,204	40%	132	5904	3	937.8	97.1 %	1.19 %
*Nostoc* sp.	PMC 881.14	8,003,053	42%	648	7711	3	845.9	99.5 %	1.19 %
*Aliinostoc* sp.	PMC 882.14	8,136,653	41%	46	7582	12	883.9	98.2 %	0.22 %
*L. boryana*	PMC 883.14	6,695,177	47%	5	6313	9	929.1	99.4 %	1.02 %
*Calothrix* sp.	PMC 884.14	13,247,652	39%	46	11,176	15	891.3	99.3 %	1.35 %
*P. couteii*	PMC 885.14	5,871,606	35%	137	5328	6	946.5	98.5 %	1.31 %

**Table 3 biomolecules-11-00028-t003:** Presence/absence of biosynthetic gene clusters of known antioxidant and/or anti-inflammatory pigments by strain. “+” and “-” indicate the detection or not of the gene clusters.

Pigments		*P. raciborskii* PMC 877.14	*Laspinema* sp. PMC 878.14	*M. vaginatus* PMC 879.14	*L. martensiana* PMC 880.14	*Nostoc* sp. PMC 881.14	*Aliinostoc* sp. PMC 882.14	*L. boryana* PMC 883.14	*Calothrix* sp. PMC 884.14	*P. couteii* PMC 885.14
Phycobiliproteins	C-Phycocyanin	+	+	+	+	+	+	+	+	+
	Allophycocyanin	+	+	+	+	+	+	+	+	+
	Phycoerythrin	+	-	-	-	+	-	-	+	+
Scytonemin		-	-	-	-	+	+	-	+	-
Hydroxylated carotenoids	Cryptoxanthin	+	+	+	+	+	+	+	+	+
	Zeaxanthin	+	+	+	+	+	+	+	+	+
	Caloxanthin	-	+	-	+	-	+	+	+	-
	Nostoxanthin	-	+	-	+	-	+	+	+	-
Myxoxanthophylls	1-hydroxylycopene	+	+	+	+	+	+	+	+	+
	1′-Hydroxy-γ-carotene	+	+	+	+	+	+	+	+	+
	Myxol 2′-pentoside	+	+	+	+	+	+	+	+	+
	2-Hydroxymyxol 2′-methylpentoside	-	+	-	+	-	+	+	+	-
Echinenone		-	+	+	+	+	+	+	+	+
Ketocarotenoids	Canthaxanthin	-	+	+	+	+	+	+	+	+
	Adonixanthin	-	+	+	+	+	+	+	+	+

**Table 4 biomolecules-11-00028-t004:** Pigment composition (mean in µg·mg^−1^ DW) of nine cyanobacteria isolated from Balaruc’s thermal mud.

Pigments		*P. raciborskii* PMC 877.14	*Laspinema* sp. PMC 878.14	*M. vaginatus* PMC 879.14	*L. martensiana* PMC 880.14	*Nostoc* sp. PMC 881.14	*Aliinostoc* sp. PMC 882.14	*L. boryana* PMC 883.14	*Calothrix* sp. PMC 884.14	*P. couteii* PMC 885.14
Carotenoids	Zeaxanthin	1.72	0.38	0.43	0.11	0.00	0.01	0.22	0.00	0.29
	Myxoxanthophyll-like	0.44	0.28	0.00	0.12	0.03	0.32	0.00	0.21	0.00
	β-Carotene	3.65	1.43	1.06	0.46	0.15	1.29	1.31	0.46	0.78
Chlorophylls	Bacteriochlorophyll a	0.54	0.04	0.12	0.04	0.06	0.08	0.07	0.07	0.02
	Chlorophyll a	35.2	9.8	11.2	4.9	1.5	12.3	17.1	9.7	6.7
	Phaeophytin a	0.00	0.09	0.07	0.02	0.01	0.00	0.09	0.04	0.00
Phycobiliproteins	Phycocyanin	32.5 ± 12.8	18.4 ± 0.2	12.2 ± 2.1	6.1 ± 0.5	2.1 ± 0.1	21.3 ± 4.7	24.9 ± 4.4	9.9 ± 0.4	3.5 ± 0.1
	Allophycocyanin	7.9 ± 3.5	6.6 ± 0.5	5.8 ± 1.1	3.7 ± 0.3	1.5 ± 0.2	15.6 ± 3.5	15.2 ± 3.8	6.7 ± 0.3	2.7 ± 0.2
	Phycoerythrin	33.3 ± 12.1	0.00	0.00	0.00	2.8 ± 0.2	0.00	0.00	9.9 ± 0.3	3.3 ± 0.1

**Table 5 biomolecules-11-00028-t005:** Specialized metabolites biosynthetic gene clusters (BGCs) from the nine cyanobacterial strains isolated from Balaruc’s thermal mud. Automatic annotations have been performed using the MicroScope platform (computing AntiSMASH and MIBiG softwares). * Some clusters have been predicted at the end/start of a contig and are questionable regarding their completeness.

Genus/Species	Strain	Nb of Clusters Predicted	Identified Clusters	Reported Activity	Unknown/Homologous Cluster	Unknown/Unknown Cluster
*P. raciborskii*	PMC 877.14	7	Phytoene or derivative	Antioxidant	1	5
*Laspinema* sp.	PMC 878.14	7	Phytoene or derivative Heterocyclic cyanobactin	Antioxidant Cytotoxic	5	0
*M. vaginatus*	PMC 879.14	7	Phytoene or derivative	Antioxidant	3	3
*L. martensiana*	PMC 880.14	9	Phytoene or derivative Squalene Schizokinen-like	Antioxidant Antioxidant Iron-chelating activity	5	1
*Nostoc* sp.	PMC 881.14	20 *	Phytoene or derivative (2×) Heterocyst glycolipid	Antioxidant n.d.	7	10
*Aliinostoc* sp.	PMC 882.14	21	Shinorine-like Microginin-like Amycomicin-like Phytoene or derivative (2×) Heterocyst glycolipid Microviridin-like	Antioxidant ACE inhibition; no cytotoxic Antibiotic Antioxidant n.d. Protease inhibitor	6	8
*L. boryana*	PMC 883.14	16	Phytoene or derivative (5×)	Antioxidant	8	3
*Calothrix* sp.	PMC 884.14	27	Cyanobactin Mycosporine-glycine-like Cylindrocyclophane-like Phytoene/Lycopene or derivative (3×) Heterocyst glycolipid	Cytotoxic Antioxidant Cytotoxic; antibacterial Antioxidant n.d.	14	6
*P. couteii*	PMC 885.14	10	Phytoene or derivative (2×)	Antioxidant	2	6
Total		124	31		51	42

**Table 6 biomolecules-11-00028-t006:** Potential antioxidant and anti-inflammatory compounds identified by HPLC-DAD or LC-MS/MS in the extracts of the nine cyanobacteria strains isolated from Balaruc’s thermal mud. MAAs: Mycosporine-like Amino Acids. “+” and “-” indicate the detection or not of the gene clusters.

Class	Compound	*P. raciborskii* PMC 877.14	*Laspinema* sp. PMC 878.14	*M. vaginatus* PMC 879.14	*L. martensiana* PMC 880.14	*Nostoc* sp. PMC 881.14	*Aliinostoc* sp. PMC 882.14	*L. boryana* PMC 883.14	*Calothrix* sp. PMC 884.14	*P. couteii* PMC 885.14
MAAs (*antioxidant*)	Mycosporine-glycine	-	-	-	-	-	+	-	+	-
	Mycosporine-ornithine	-	-	-	-	+	-	-	-	-
	Nostoc-756	-	-	-	-	+	-	-	-	-
	Shinorine	-	-	-	-	-	+	-	-	-
Carotenoids (*antioxidant*)	Canthaxanthin	-	-	+	+	-	+	+	+	+
	Myxoxanthophyll-like	+	+	-	+	-	+	-	+	-
	Zeaxanthin	+	+	+	+	-	-	+	-	+
	β-carotene	+	+	+	+	+	+	+	+	+
Chlorophylls (*antioxidant*)	Chlorophyll a	+	+	+	+	+	+	+	+	+
	Bacteriochlorophyll a	+	-	+	-	-	-	-	-	-
Phycobiliproteins (*antioxidant*)	Phycocyanin (*Anti-inflammatory*)	+	+	+	+	+	+	+	+	+
	Allophycocyanin	+	+	+	+	+	+	+	+	+
	Phycoerythrin	+	-	-	-	+	-	-	+	+
	Phycoerythrobilin	-	-	-	-	-	-	-	+	+
Aeruginosins (*anti-inflammatory*)	Aeruginosin TR642	-	-	-	-	-	-	+	-	-
	*Antioxidant*	8	6	7	7	7	8	6	9	8
	*Anti-inflammatory*	1	1	1	1	1	1	2	1	1

**Table 7 biomolecules-11-00028-t007:** Antioxidant, anti-inflammatory, and wound-healing activities detected for methanol and aqueous extracts of Balaruc cyanobacteria. - represents no significant effect; - x%, significant inhibiting effect percentage; + x%, significant increasing effect percentage.

			*P. raciborskii* PMC 877.14		*Laspinema* sp. PMC 878.14		*M. vaginatus* PMC 879.14		*L. martensiana* PMC 880.14		*Nostoc* sp. PMC 881.14		*Aliinostoc* sp. PMC 882.14		*L. boryana* PMC 883.14		*Calothrix* sp. PMC 884.14		*P. couteii* PMC 885.14	
Bioassay		µg·mL^−1^	MeOH	H_2_O	MeOH	H_2_O	MeOH	H_2_O	MeOH	H_2_O	MeOH	H_2_O	MeOH	H_2_O	MeOH	H_2_O	MeOH	H_2_O	MeOH	H_2_O
**NO secretion** RAW 264.7 (*antioxidant*)		1	-		-		-		-		-		-		-		-		-	
		5	-		-		-		-		-		-		-		-		-	
		10	-		-		-		-		-		-		-		-		-	
		50	-		-		-		-		-	+20 ± 2	-		-		-		-	
**Keratinocytes migration** HaCaT (*wound healing*)		1	-		-		-		-		-		-		-		-		-	
		5	-		-		-		-		-		-		-		-		-	
		10	-		-		-		-		-		-	+22 ± 7	-		-		-	
		50	-		-		-		-		-		-		-		-		-	
**Cytokines secretion** PBMC (*anti-inflammatory*)	TNF α	1	-		-		-		-		-		-		-		-		-	
	IL-1β		-		-		-		-		-		-		-		-		-	
	IL-6		-		-		-		-		-		-		-		-		-	
	IL-8		-		-		-		-		-		-		-		-		-	
	TNF α	5	-		-		-		-		-		-		-		-		-	
	IL-1β		-	−20 ± 5	-		-	+34 ± 3	-		-	+50 ± 8	-	+26 ± 3	-		-		-	
	IL-6		-		-		-		-		-		-		-		-	−27 ± 7	-	
	IL-8		-		-		-		-		-		-		-		-		-	
	TNF α	10	-		-		-		-		−18 ± 4	−20 ± 4	-		-		-		-	
	IL-1β		-	−28 ± 2	-		+23 ± 4	+49 ± 6	-		−26 ± 4	+52 ± 7	-	+30 ± 3	-		-	+31 ± 2	-	
	IL-6		-		-		-		-		-		-		-		-	−27 ± 4	-	
	IL-8		-		-		-		-		-		-		-		-		-	
	TNF α	50	−19 ± 6	-	-		-		−20 ± 1	-	-		-		-		-		-	
	IL-1β		−39 ± 9	−57 ± 3	-		+29 ± 3	+61 ± 2	-	+18 ± 3	−45 ± 5	+60 ± 4	-	+26 ± 5	-		-	+20 ± 2	-	−31 ± 2
	IL-6		-		-		−26 ± 5	-	-		-		-	−23 ± 4	-		-	−29 ± 3	-	−24 ± 3
	IL-8		-		-		-		-		-		-		-		-		-	

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
