# Peer review of "Anti-Inflammatory, Antioxidant, and Wound-Healing Properties of Cyanobacteria from Thermal Mud of Balaruc-Les-Bains, France: A Multi-Approach Study"

_biomolecules, 2020, doi:10.3390/biom11010028_

Round 1

Reviewer 1 Report

The authors presented a paper describing the Anti-inflammatory, antioxidant, and wound healing properties of cyanobacteria from thermal mud of Balaruc-les-Bains. To obtain the results, the authors carried out a multi-approach study, from whole-genome sequencing to metabolomics analysis, using chromatographic techniques associated with mass spectroscopy.

In addition to the clarity of the methodologies adopted, described in detail, the scheme of the manuscript allows for a fluent and clear follow-up, and the results here presented highlight the beneficial effects of the thermal mud of Baraluc-les-Bains.

Personally, I have no particular criticisms to point out to the authors. Just a few things I would like to ask the authors.

- The discussion essentially focuses on the results obtained, while it does not offer ideas for further lines of research and insights. For example, a discussion on the possible anticancer or antibacterial properties of metabolites produced by the species would be useful.

- Paragraph 2.1 Sampling and taxonomy should be described in this manuscript, as the reference "Duval et al" is not reported in the list of references

- Paragraph 2.7 The methodology of the quantification of NO should be detailed in this work.

- Most of the supplementary data was not included in the submission and therefore not evaluated

- References to tables and/or figures are incorrect and indicated as "Error! Reference source not found."

- It is probable that the metabolites produced by the flora as a whole in the thermal mud may differ from those produced individually by the species analyzed. This could explain the lack of some of the properties expected by the authors. A brief discussion of this aspect could enrich the manuscript.

Author Response

Thank you for your revisions,

Reviewer 2 Report

This interesting study is aimed to obtain genomic and metabolomic information about nine different cyanobacterial strains that have been isolated from the Balaruc-les-Bains’ thermal mud. Considering the importance of cyanobacteria for the search of new biologically active compounds, the presented study is relevant.

To help to improve the perception of the text of the manuscript by readers, it is necessary to make some additions to the text.

Below are the comments of the reviewer.

  • Genome sequencing information must be deposited into the GenBank. The GenBank information (the GenBank accession numbers) has to be presented in the manuscript.
  • GenBank information about 16S rRNA sequences of the nine new strains has to be shown in this manuscript. These sequences were used for phylogeny analysis (Figure S1).
  • Please, indicate the cultures age when you collected them for metabolite extraction.
  • Authors have to keep in mind that metabolite synthesis depends on the growth stage and growth conditions. It is important to remember also that while cyanotoxins are absent in laboratory growth conditions, but they can appear in natural conditions.
  • Figure 1 is difficult for readers understanding. Figures details are very tiny. It is necessary to explain this Figure in the text. The same comments can be applied to the Figure S2.
  • Please, provide full statistical information. How many biological replicates and technical replicates were made in this study? For example, in case of the Table 4, please, show deviations. How many repetitions were performed?

Author Response

Thank you for your revisions,

Please find our responses attached.

Author Response

Thank you for your review, please find our responses attached.

Round 2

Reviewer 2 Report

The authors answered the reviewer's questions and made edits to the manuscript.